# How personal values shape job seeker preference: A policy capturing study

**Carol L. Hicklenton**[1]*, **Donald W. Hine**[2], **Aaron B. Driver**[3], **Natasha M. Loi**[1]

**1** School of Psychology, University of New England, Armidale, Australia, **2** School of Psychology, Speech and Hearing, University of Canterbury, Christchurch, New Zealand, **3** UNE Business School, Armidale, Australia

* carol.hicklenton@gmail.com

**Data Availability Statement:** All work climate, personal values, and organization attractiveness files are available from the RUNE database (accession number(s) 10.25952/5fffc354212c6).

## Abstract

Does the "ideal" organization exist? Or do different workplace attributes attract different people? And if so, what attributes attract what types of employees? This study combines person-organization fit theory and a policy capturing methodology to determine (a) which attributes are the strongest predictors of perceived organization attractiveness in a sample of Australian job seekers, and (b) whether the magnitude of these predictive effects varies as a function of job seekers' personal values. The design of this study is a randomized experiment of Australian job seekers who responded to an online survey invitation. Each of the 400 respondents received a random subset of 8 of 64 possible descriptions of organizations. Each description presented an organization that scored either high or low on six attributes based on the Employer Attractiveness Scale: economic, development, interest, social, application, and environmental value. Multi-level modelling revealed that all six attributes positively predicted job seekers' ratings of organization attractiveness, with the three strongest predictors being social, environmental, and application value. Moderation analyses revealed that participants with strong self-transcendent or weak self-enhancement values were most sensitive to the absence of social, environmental, and application value in workplaces, down-rating organizations that scored low on these attributes. Our results demonstrate how job seekers' personal values shape preferences for different types of workplaces. Organizations may be able to improve recruitment outcomes by matching working conditions to the personal values of workers they hope to employ.

## Introduction

Securing high-quality employees is critical to the success of business organizations. Successful recruiting involves not only being judged as attractive by desirable job applicants, but also being the employer-of-choice for applicants weighing several offers. But what exactly makes organizations attractive to potential applicants? Industry leaders, such as Google and Apple, can use name recognition and reputation to attract desirable applicants, but other less high-profile organizations must rely on alternative strategies. Research into the attractiveness of organizational attributes varies substantially across studies. Some studies indicate that factors

**Funding:** The authors received no specific funding for this work.

**Competing interests:** The authors have declared that no competing interests exist.

like pay and promotion potential are most important, whereas others highlight the prospect of challenging and interesting work, opportunities for teamwork and positive social interactions [1, 2]. Person-organization (PO) fit, operationalized in this study as the level of congruence between values of organizations and their prospective employees, is a useful theoretical lens for understanding these inconsistencies [3]. This study combines PO fit theory and a policy capturing methodology to determine which organizational attributes are the strongest predictors of perceived organization attractiveness in a sample of Australian job seekers, and whether the magnitude of these predictive effects varies as function of job seekers' personal values.

## Organizational attractiveness

Organizational attractiveness refers to the overall appeal of an organization to employees, prospective employees, and others who may choose to engage (or not engage) with it. Attractiveness can be conceptualized as an expectancy of "envisioned benefits" and/or as "an attitude or expressed general positive affect", reflecting the general desirability of initiating or maintaining a relationship with a particular organization [4–6].

To date, two meta-analyses have summarized much of the research on workplace attributes and organization attractiveness [1, 2]. The first review of 71 studies, conducted by Chapman et al., found that work environment and organization image (reputation) were much stronger predictors of perceived organization attractiveness ($r = .60$ and $.48$, respectively) than job characteristics such as pay ($r = .27$) and promotion opportunity ($r = .27$) [1]. A subsequent review of 232 studies by Uggerslev et al. also found organization image to be a stronger predictor of perceived organization attractiveness ($r = .48$) than pay ($r = .23$) and promotion opportunity ($r = .35$) [2]. Interestingly, work environment, the strongest predictor in the first meta-analysis, was more modest in the second ($r = .30$) [2].

Both meta-analyses reported statistically significant Q coefficients for most predictors, reflecting heterogeneity in effect sizes across studies. That is, the effects of specific workplace attributes on perceived organization attractiveness varied significantly across studies; different studies often identified different workplace attributes as the primary drivers of participants' perceptions of organization attractiveness. Moderation analyses conducted in both reviews examined whether average effect sizes for a given attribute varied as a function of sample characteristics such as gender and nationality, and examined measurement approaches for assessing organization attractiveness or organization attributes. For example, Chapman et al. found that women placed more weight on job characteristics such as location and pay than did men, and job applicants were likely to weigh justice perceptions more strongly than non-applicants [1].

Heterogeneous effects have also been identified within, as opposed to across, studies. Alnıaçık et al. compared the mean attractiveness scores for 25 organization PO attributes across two nationalities and reported significant cross-national differences for 24 of the 25 attributes [7]. An above-average basic salary was the only attribute for which there was no significant difference in attractiveness rating by nationality.

Many studies investigating the associations between workplace attributes and job seekers' perceptions of organization attractiveness have employed ad hoc strategies, focusing on one or a few individual attributes making it difficult for business organizations to use research findings for guiding their recruitment strategies. In an attempt to develop a more systematic and comprehensive framework for assessing organization attributes that predict attraction, Berthon et al. developed the Employer Attractiveness Scale (EAS) [4]. Benefits of the EAS include a structure derived from both interviews and a factor analysis, with item descriptions that encompass a broad range of work values. The work values in the EAS are categorized from the

perspective of potential job applicants, and align with psychological processes that might explain perceptions of attractiveness. The 25 items in the EAS assess five dimensions: (1) *economic value* (basic salary, overall compensation, job security and promotion opportunities); (2) *development value* (supporting employees' personal and career development); (3) *interest value* (supporting novel work practices and forward thinking, and valuing and making use of employees' creativity in the production of high quality and innovative products and services); (4) *social value* (providing a positive and pleasant social environment for employees); and (5) *application value* (being humanitarian and customer-oriented, and providing opportunities for employees to apply their knowledge, teach others, and experience acceptance and belonging).

One important limitation of the EAS is that its *application value* dimension does not include alternative types of corporate social responsibility (CSR), such as a commitment to sustainability and positive environmental outcomes. Corporate environmental responsibility is sometimes regarded as a sub-type of CSR [8, 9]. In an influential review, Orlitzky et al. suggested that researchers should focus on specific sub-dimensions of CSR, and several studies have found that corporate environmental responsibility is an important predictor of organization attractiveness [5, 8, 10–12]. Given these findings, in the current study we employ an expanded EAS framework which includes both application value (reflecting CSR) and environmental value (reflecting corporate environmental responsibility).

## Person-organization fit and perceptions of attractiveness

PO fit provides a useful conceptual framework for investigating personal values and perceptions of organization attractiveness, and for understanding why certain workplace attributes are strong predictors of perceived organizational attractiveness in some studies but not others [3]. PO fit is broadly defined as the compatibility between individuals and organizations [3]. Compatibility is conceptualized as complementary fit and supplementary fit. Complementary fit occurs when a "weakness or need of the environment is offset by the strength of the individual or vice versa" [13]. Supplementary fit refers to situations where the person and the organization possess similar characteristics, such as when work values promoted by recruiting organizations match personal values of potential job applicants [3]. The present study focuses on supplementary fit between work values and personal values as they relate to job-seekers' perceptions of organization attractiveness. PO fit can help explain that attraction is not based on organization attributes per se but on how those attributes match employees' or prospective employees' values, skills, and interests. To date, most of the research on PO fit has employed perception-based measures where respondents are asked how well organizations fit their values and needs (e.g., "To what degree do you feel your values 'match' or fit this employer?") [14].

In the present study, we employed an alternative approach. Job seekers were asked to evaluate the attractiveness of a range of organizations that either provide weak or strong support for a range of workplace outcomes (i.e., economic, development, interest, social, application, and environmental outcomes). Using a policy capturing methodology, we evaluated whether job seekers would focus on different features of organizations when generating their attractiveness judgements. In this approach, increased PO fit is reflected in the degree to which an organization's support for specific workplace outcomes matches job seekers' personal values.

## Personal values and perceived organization attractiveness

Schwartz's value theory provides a useful conceptual model for understanding precisely why job seekers with different values would prefer certain types of organizations more than others

[15, 16]. According to Schwartz, personal values reflect desired goals that apply in a broad range of situations, and implicitly or explicitly serve as guiding principles in people's work and personal lives [17]. Schwartz's model is most commonly presented as a circumplex with nine value dimensions: self-direction, universalism, benevolence, conformity, security, power, achievement, hedonism, and stimulation [18]. The values captured by Schwartz's circumplex are often combined into sets of superordinate values [18]. Of relevance to this study are two superordinate values: self-transcendence and self-enhancement. These are sometimes referred to as "other" and "self" orientations [19].

Self-transcendent values "emphasize concern for the welfare and interests of others" and encompass universalism and benevolence [18]. Universalist values derive from "the survival needs of individuals and groups" which contrasts with the in-group focus of benevolence values. Benevolence values are defined by goals of "preserving and enhancing the welfare of those with whom one is in frequent personal contact" [18]. Self-enhancement value types, on the other hand, prioritize achievement and power. Achievement is defined as "competent performance that generates resources". Power is defined as "control or dominance over people and resources" [18].

Schwartz argues that people tend to endorse all personal values to some degree, but prioritize them differently [17]. The process of value prioritization makes certain organization attributes more personally relevant to job seekers than others. For example, someone who prioritises self-enhancement might rate organizations that provide employees with generous financial remuneration and opportunities for training advancement as more attractive than organizations that do not. Alternatively, a job seeker who prioritises self-transcendence might be more attracted to organizations with a strong commitment to CSR.

To date, several studies have assessed how personal values predict job seekers' and current employees' perceptions of organization attractiveness. In an early study, Cable and Judge hypothesized that certain types of pay systems would be generally preferred over others, and that different types of job seekers would prefer different pay systems [20]. They found that job seekers, overall, preferred organizations that offered high pay, flexible benefits, pay based on individual performance, and fixed pay that was not contingent on the overall performance of the organization. Consistent with the pay-person fit hypothesis, Cable and Judge also found that job seekers with stronger materialist values were particularly attracted by high pay levels, whereas those with stronger collectivist values were more opposed to pay systems that rewarded individual as opposed to group performance [20].

More recently, Bridoux et al. conducted a study assessing the trade-offs stakeholders are willing to make when deciding to associate with a firm (e.g., by purchasing a product from the firm or seeking employment there) [19]. They found that stakeholders who scored higher on self-transcendent values were more willing to trade-off personal material benefits to secure improved conditions for suppliers from developing nations. In contrast, stakeholders with stronger self-enhancement values were more attracted to firms that favoured their own in-group. The current study extends previous research by examining the interplay between a much broader range of workplace attributes and personal values on perceptions of organization attractiveness.

## The current study

This study employed a policy capturing methodology to determine which workplace attributes are the most important drivers of perceived attractiveness of organizations in a sample of Australian job seekers. Utilizing PO fit theory and multi-level modelling, it also evaluated whether the magnitude of predictive effects varied as a function of job seekers' values [3, 21]. Although

previous studies have examined PO fit using a policy capturing methodology, the present study is unique for a number of reasons [20, 22]. First, whereas, these PO fit policy capturing studies relied on small university student samples, we employed a larger, more representative sample of employed Australians who were planning to switch jobs within the next year. Second, whereas previous research focused on an ad hoc collection of one or a few predictors of organization attractiveness, our study used a more comprehensive EAS-derived framework that assessed six workplace dimensions: economic, development, interest, social, application, and environmental value [4]. Our research is the first policy capturing study to investigate whether job seekers with different personal values focus on different EAS dimensions when constructing their judgements of organization attractiveness.

Based on research conducted using the EAS and meta-analyses by Chapman et al. and Uggerslev et al., we predicted that organizations that support positive workplace outcomes related to economic, development, interest, social, application, and environmental values would be more attractive than organizations that do not support these values (Hypothesis 1) [1, 2, 4]. Based on the effect sizes presented in the meta-analyses, we predicted that organization support for positive employee relations (social value) would be a particularly strong driver of perceived attractiveness (Hypothesis 2). Challenging and interesting work (interest value), personal and career development (development value), and pay and promotion opportunities (economic value), would be reliable but weaker predictors of attractiveness than social value (Hypothesis 3). Given previous findings that commitment to positive societal and environmental outcomes are stronger predictors of perceived organization attractiveness than promotion opportunities and pay, we predicted that the attractiveness of both application value (Hypothesis 4) and environmental value (Hypothesis 5) would be stronger than economic value, interest value, and development value [5, 12].

Based on PO fit theory and research, we predicted that job seekers' self-transcendent and self-enhancement values would moderate the predictive effects of workplace attributes on perceived attractiveness [23]. Specifically, organizations with strong commitment to supporting social, application, and environmental outcomes would be perceived as more attractive by job seekers with stronger self-transcendent values, relative to those with weaker self-transcendent values (Hypothesis 6). Organizations committed to supporting positive economic, interest, and development outcomes would be perceived as more attractive to job seekers with stronger self-enhancement values relative to those with weaker self-enhancement values (Hypothesis 7).

## Methods

### Participants

A sample of 400 Australian adults, recruited from a Qualtrics™ (Provo, UT) online panel, participated in this study. At the time of recruitment, all participants indicated that they were employed full-time but looking to change jobs within the next 12 months (assessed by a screening question at the beginning of the survey). Close to two-thirds of participants were women (62%), somewhat higher than the Australian workforce participation rate of 54.8% [24]. Ages ranged from 19 to 75 years: 18–24 (6%), 25–34 (19.5%), 35–44 (30%), 45–54 (23%), 55–64 (16%), and 65 to 75 years (5.5%). The mean age of our sample of 42 years was slightly higher than the national average of 38 years [24]. The sample included a broad range of education levels: less than Year 10 (<1%), Year 10 high school (4%), Year 12 high school (10%), vocational education training certificate (15%), diploma or advanced diploma (13%), graduate diploma or bachelor degree (43%), and postgraduate university degree (16%). University graduates were somewhat over-represented in our sample compared with the national average in which 34% of the labour force has a degree or higher [25].

## Procedure

The survey was developed and delivered using the Qualtrics™ online survey platform. Prior to data collection, the host institution's Human Research Ethics Committee reviewed and approved the project. The surveys were administered between 26 June and 13 July, 2017. A screening question preceding the survey was used for participant consent. Participants indicated their agreement with written survey information by clicking a button. All participants indicated being over 18 years of age and received payment under $3, administered by Qualtrics™, after completing the survey. The first part of the survey, immediately following the screening question, assessed demographics and personal value orientations. Each participant then read a random selection eight descriptions of organizations (selected from 64 in total), which varied all possible combinations of six attributes relevant to job search (e.g., salary, opportunities for career development, environmental policies, etc.). Effective policy capturing design requires enough scenarios and cues for stable estimates, but not too many for respondents to become bored or fatigued [5]. For continuity, the six attributes were presented in the same order in each organization description. After reading each description, participants completed five items assessing how attracted they were to the organization as a potential employer. A central aim of the study was to assess whether certain types of work environments would be perceived as more attractive, depending on participants' values. Details about the experimental stimuli and measures are presented below.

## Manipulations and measures

**Organization attributes.** Based on the EAS, we created 64 unique descriptions of organizations that varied on six dichotomous attributes reflecting the degree to which the organization: (a) provided a good salary and promotion opportunities (economic value); (b) supported employees' personal and career development (development value); (c) possessed a reputation for being exciting and innovative, encouraging creativity, and providing a challenging work environment (interest value); (d) provided a positive and pleasant social environment for employees (social value); (e) exhibited a strong commitment to customer focus, social and racial equality, and operating in a manner that supports society (application value); and (f) had strong pro-environmental policies and procedures, and encouraged environmentally sustainable practices (environmental value) [4].

The first five attributes were based on five facets of the EAS (4). Environmental value was a new attribute developed for this study to assess the degree to which prospective job applicants value organizations' commitment to environmental sustainability when considering employment options. The organization descriptions reflected all possible combinations of the attributes, ensuring that the attributes were all orthogonal. A summary of the high and low descriptors for each organizational attribute is presented in S1 Table.

Scenarios were created by combining the text presented in S1 Table in all possible combinations. No additional text was added, and all scenarios were constructed with text blocks in the same order (i.e., economic value first, followed development, interest, social, application, and environmental value).

**Organization attractiveness.** Following the presentation of each description, organization attractiveness was assessed as the extent to which participants felt attracted to the organization and intended to pursue employment with that organization. Using multi-level modelling, participants' reactions to each organization description were assessed as organization attraction and job pursuit intention with five items used by Aiman-Smith et al. [5, 21]. Representative items include, "This would be a good company to work for" and "I would like to work for this company" for organization attraction, and "I would actively pursue obtaining

a position with this company" and "I would accept a job offer from this company" for job pursuit. All responses were measured using a 7-point scale (1 = *very unlikely*, 7 = *very likely*). An overall attractiveness scale was computed by taking the mean of all attraction and job-pursuit items. Cronbach's alpha for the scale was .98.

**Self-transcendence and self-enhancement values.**  Participants' personal values were assessed by the self-transcendence (15 items) and self-enhancement (9 items) subscales of the most recent version of the Portrait Values Questionnaire [16]. All items were comprised of brief, gender-matched portraits portraying the motivation or aspirations of a fictitious person (e.g., "It is important to her to care for nature" and "It is important to her to be wealthy"). Participants rated how similar they are to the person portrayed in the portrait on a 6-point scale (1 = *not like me at all*, 6 = *very much like me*). Scores for self-transcendent and self-enhancement value orientations were computed by taking the mean of relevant items identified by Schwartz [18]. Self-transcendence was computed by taking the mean scores across 15 items assessing universalism and benevolence (α = .88), and self-enhancement was computed as the mean of nine items assessing power and achievement (α = .87).

## Results

### Descriptive statistics

Means, standard deviations, and intercorrelations for organization attractiveness, self-enhancement, and self-transcendent values were examined using SPSS V25. On average, participants reported they were moderately attracted to organization attributes as presented in the scenarios with the mean on the organization attractiveness measure falling above the midpoint (4.22 on a 1 to 7 scale, $SD$ = 1.78). The sample mean on the self-enhancement scale (3.38, $SD$ = .93) fell just below the midpoint on the 1 to 7-point scale, and the mean for self-transcendence (4.77, $SD$ = .63) fell above the midpoint. Self-enhancement and self-transcendent values were weakly correlated ($r$ = .09, $p$ = .07).

**Workplace attributes predicting job seekers' perceptions of organization attractiveness.**  We used policy capturing and multilevel modelling to test our hypotheses. Policy capturing is a method used in applied psychology to investigate the associations between people's judgements and cues in the environment used to make those judgements [26]. The present study explored which workplace attributes job seekers use when constructing judgements about the attractiveness of organizations as potential employers. Multilevel modelling is a highly flexible, regression-based statistical strategy for quantifying the magnitude of the relationship between environmental cues and judgements. It is specifically designed to analyse data with hierarchical or nested structures. Given that each participant in the study provided attractiveness judgements for eight hypothetical workplaces, participants' judgements (level of organization attraction and job pursuit intention) were nested within their reactions to each organization description presented. In the Level 1 (within-person) analysis, regression equations were created for each participant using attractiveness as the outcome variable and the six organization attributes from the scenarios as predictors (i.e., economic, development, interest, workplace, application, and environmental values). This enabled the study to determine which organization attributes predicted higher ratings of perceived attractiveness. Each of the organization attributes were coded 1 for the low condition and 2 for the high condition.

The Level 2 (between-person) analysis involved regressing the intercepts and beta coefficients from the Level 1 analysis on participants' scores on self-enhancement and self-transcendent values. The Level 2 analysis assessed whether the relations between organization attributes and attractiveness decisions varied systematically as a function of pre-existing personal values. In this study, all policy capturing analyses were conducted using HLM 6, using

restricted maximum likelihood estimates and robust standard errors [21]. For all analyses, a conservative α cut-off of $p < .01$ was adopted.

Sample size is an important consideration in multilevel studies. Maas and Hox [27] ran a series of simulations in which they varied Level 1 and Level 2 sample sizes. They found that only simulations with small Level 2 samples (consisting of 50 or fewer cases) produced biased Level 2 standard errors. All other simulations, including those with Level 1 sample sizes as small as five, produced accurate and unbiased regression coefficients, variance components, and standard errors at both Level 1 and Level 2. Given the current study had 400 participants at Level 2 and six ratings per respondent at Level 1, it exceeded Maas and Hox's recommended sample size guidelines [27].

**Unconditional model.** An initial unconditional model (i.e., no predictors at within-person or between-person levels) divided the total variance in organization attractiveness judgments into within- and between-person components. The intraclass correlation computed from the unconditional model was.40, indicating that 40% of the variance in attractiveness was attributable to individual differences (between-subjects variance). In other words, irrespective of the specific workplace attributes presented in the organization descriptions, substantial variation in perceived attractiveness judgements across participants was evident. The remaining 60% of the variance in the data set reflected within-subjects variance across the six attributes, indicating substantial variation within participants depending on the specific array of workplace attributes in each organization description. Given that the intraclass correlation was large, multi-level analysis was an appropriate strategy [28].

**Level 1 model: Which workplace attributes predict organization attractiveness?** The Level 1 analysis involved regressing organization attractiveness (the criterion variable) on six dichotomous predictors reflecting low or high economic, development, interest, social, application, and environmental value. Average unstandardized coefficients and robust standard errors for the intercept and each of the workplace attributes are presented in S2 Table.

The intercept value of 4.22 ($SE = .05$) indicates that, on average, participants' organization attractiveness judgements fell just above the midpoint on the 6-point scale. All six attributes significantly predicted participants' attractiveness judgements, with social, environmental, and application value being the three strongest predictors. That is, providing positive social environments, strong environmental policies and practices, and a commitment to customer and societal welfare were the strongest drivers of job seekers' judgements of organization attractiveness.

**Level 2 model: Do personal values moderate the effects of workplace attributes on perceived attractiveness?** A major aim of this study was to determine whether the degree to which workplace attributes predict perceived organization attractiveness varies as a function of job seekers' personal values. As previously stated, the PO fit hypothesis suggests that organizations with work environments that match workers' personal values should be perceived as more attractive. To address this, we conducted a Level 2 analysis in which the strength of job seekers' self-enhancement and self-transcendent values were used to predict the intercepts and beta coefficients associated with each of the organization attributes from the Level 1 analysis. Significant Level 2 effects are referred to as cross-level interactions because they indicate the magnitude of the relations between the Level 1 predictors (workplace attributes) and the criterion (perceived organization attractiveness). Significant cross-level indirect effects show the extent that perceptions of attractiveness vary as a function of the value of Level 2 predictors (personal values). To aid in the interpretation of cross-level interactions, all significant Level 2 effects were plotted using HLM's graph module. A summary of the Level 2 analysis is presented in S3 Table.

Plots of the significant interactions are presented in S1 Fig. Plots of the nonsignificant interactions are presented in S2 Fig.

Examination of the intercept analyses indicated that participants with higher self-enhancement values were significantly more attracted to the organizations described in the study than participants with lower self-enhancement values. The opposite pattern was evident for self-transcendence. Participants with higher self-transcendence values were less attracted to the organizations, overall, than participants with low self-transcendence values, although this effect just failed to reach statistical significance ($p$ = .01).

Examination of the cross-level interactions revealed significant effects between personal values and workplace attributes related to social, application, and environmental value.

All the interactions involving self-transcendence followed the same general pattern (see S1 Fig, top row). Organizations with high social, application, and environmental value were perceived as being highly attractive by all participants (irrespective of whether they had weak or strong self-transcendence values), with very little differentiation between the two groups. However, participants with high self-transcendence values were more sensitive to the absence of these three organizational attributes. Consistent with the PO fit hypothesis, high "self-transcenders", relative to low "self-transcenders", rated organizations that scored low on these attributes as much less attractive.

Interactions involving self-enhancement took a slightly different form than for self-transcendence. Overall, organizations, regardless of whether they had high or low social, application, and environmental value, were perceived as more attractive by participants with strong self-enhancement values than those with weak self-enhancement values. However, high self-enhancers were less sensitive to the absence of these three attributes than low self-enhancers. That is, whereas high self-enhancers perceived organizations with low social, application, and environmental value to be only somewhat less attractive compared to organizations that scored high on these attributes, low self-enhancers perceived organizations with low social and application value as significantly less attractive. Low self-enhancers also perceived organizations with low environmental value as less attractive, although this interaction effect failed to reach statistical significance ($p$ = .06).

## Discussion

This study investigated which workplace attributes most strongly predict perceptions of organization attractiveness in a sample of Australian job seekers, and whether the magnitude of these predictive effects vary as a function of job seekers' personal values. We found that workplaces with attributes reflecting higher levels of economic, development, interest, social, application, and environmental value were perceived as more attractive than workplaces lacking these attributes. We also found that the strength of the predictive effects for social, application, and environmental value varied as a function of job seekers' personal values. This finding is broadly consistent with PO fit theory, which suggests that matches between workplace attributes and job seekers' personal values should produce higher ratings of perceived organization attractiveness [3]. These findings are explored in more detail in the sections that follow, along with comments regarding the limitations of the study and recommendations for future research.

### Which workplace attributes are the strongest predictors of organization attractiveness?

When job seekers decide to apply or not apply for a job, they often do so based on how well potential workplaces stack up on key dimensions related to remuneration, CSR, intellectual

stimulation, and so on. A central aim of this study was to determine which of six workplace attributes, based on the EAS, are the primary drivers of job seekers' perceptions of organization attractiveness [4]. Consistent with Hypothesis 1, organizations that support positive workplace outcomes related to economic, development, interest, social, application, and environmental values were judged as more attractive than organizations that do not support these values. Also as hypothesized, the strongest predictors of perceived attractiveness were social value (providing positive social environments; Hypothesis 2), application value (commitment to customer and societal welfare; Hypothesis 4), and environmental value (strong environmental policies and practices; Hypothesis 5), all of which were significantly stronger predictors of perceived organization attractiveness than economic value (pay and promotions), development value (supporting personal and career development), and interest value (provision of challenging and interesting work), providing support for Hypothesis 3. Of all the workplace attributes evaluated as part of this study, social value was by far the strongest predictor of perceived organization attractiveness, with a coefficient more than twice the size of the next highest predictor, environmental value. This result is consistent with Uggerslev et al.'s meta-analysis which also found positive employee relations and treatment to be the strongest predictor of attractiveness [2].

In the current study, environmental value (reflecting corporate environmental responsibility) was the next strongest predictor of attractiveness, followed by application value (reflecting CSR). Beta coefficients for environmental and application value, while roughly half the size of social value, were each about three times larger than the predictive effects for economic, development, and interest value. These findings support previous research highlighting the importance of having highly visible corporate social and environmental responsibility strategies [11, 12]. Not only are these strategies good for society and the environment, they are also attractive to prospective employees and customers.

Our findings also support previous research which suggest that while good pay and promotion opportunities are significant predictors of perceived organization attractiveness, their effect sizes are modest [1, 2]. This finding supports the meta-analysis by Uggerslev et al. in which pay ($r = .23$), promotion ($r = .35$), and development ($r = .49$) each were statistically reliable, though not particularly strong predictors of attractiveness [2]. Our results also mirror the overall pattern of findings in Uggerslev et al.'s meta-analysis, in which the effect size for training and development opportunities (development value) was stronger than for challenging and stimulating work environments (interest value), which in turn was stronger than pay (economic value) [2]. When job seekers evaluate prospective employers, pay rates are important but they are not the predominant driver of attractiveness judgements. Other factors such as providing positive social environments and commitments to corporate social and environmental responsibility appear to be much more important.

## Fit between workplace attributes and job seekers' personal values

A major aim of the study was to determine whether the effects of specific workplace attributes on perceived organization attractiveness would vary as a function of job seekers' personal values. Based on PO fit theory, we hypothesized that organizations would be perceived as particularly attractive when workplace attributes matched job seekers' personal values [3]. More specifically, we predicted organizations with strong commitment to supporting social, application, and environmental outcomes would be perceived as more attractive by job seekers with stronger self-transcendent values relative to those with weaker self-transcendent values (Hypothesis 6). Organizations committed to supporting positive economic, development, and interest outcomes would be perceived as more attractive to job seekers with

stronger self-enhancement values relative to those with weaker self-enhancement values (Hypothesis 7).

Our results only partially supported these hypotheses. With respect to Hypothesis 6, we found that organizations providing high social, application, and environmental value were perceived to be quite highly attractive for all respondents regardless of their weak or strong self-transcendent value orientation. However, differences emerged when workplace attributes related to social, application, and environmental value were absent. Job seekers with stronger self-transcendent values were more sensitive to the absence of these attributes, judging organizations without these attributes as much less attractive than organizations that had them. This suggests that when it comes to PO fit, the absence of key attributes that job seekers value may be a more important determinant of decisions not to pursue a specific job than the presence of workplace attributes they do not value. For example, job seekers who score low on self-transcendence would still find attractive a workplace that fosters strong positive social ties and supports corporate social and environmental responsibility even if they do not highly value these attributes. However, a job seeker who values these same workplace attributes would find their absence to be off-putting and potentially intolerable.

Our findings on PO misfit support previous research. For example, the study on pay preferences by Cable and Judge found that among all the interactions, the strongest interaction, reflecting PO misfit, was for collectivism and individual pay ($r = -.37$) [20]. This negative relationship was much stronger than the next closest interaction, which was a positive interaction for risk aversion and fixed pay ($r = .27$) reflecting PO fit.

In terms of Hypothesis 7, we found no evidence to support our prediction that job seekers with stronger self-enhancement values, relative to those with weaker self-enhancement values, would perceive organizations committed to strong economic, development, and interest values to be more attractive. No significant cross-over interactions between self-enhancement values and these three workplace attributes were present, indicating that organizations that provided high economic, development, and interest value (relative to those that did not) were perceived as more attractive to all respondents regardless of their value orientations. However, as noted earlier, it is important to acknowledge these effects were modest in magnitude.

Although not included in our a priori hypotheses, we did find significant cross-over interactions between self-enhancement values and workplace attributes related to positive social environments and corporate social and environmental responsibility (i.e., social, application, and environmental value, in EAS terminology). These interactions indicated that organizations with high or low social, application, and environmental attributes were perceived as more attractive by participants with stronger self-enhancement values than those with weaker self-enhancement values. However, low self-enhancers were more sensitive than high self-enhancers to the absence of social, environmental, and application value, rating organizations that scored low on social and application value as being significantly less attractive and low environmental value narrowly missing significance ($p > .01$). Our results support Bridoux et al. who found that individuals with a high other orientation (i.e., self-transcendent values) were more likely to want to associate with an organization that displayed high CSR [19].

## Practical implications

The current study provides general guidance to businesses about how to increase the attractiveness of their public profile and brand. Many organizations already have such initiatives in place. However, for the most part, they mainly target potential consumers or investors; initiatives aimed at potential employees are less common. Our results indicate that highlighting corporate social and environmental responsibility, and supportive collegial working

environments, may also be an important tool for recruiting participants. For example, a multinational manufacturer of boots, shoes, and clothing, claims an "unwavering commitment to environmental and social responsibility" and pays its employees to volunteer on projects for "underserved communities" such as Urban-Greening-Los-Angeles [29–31]. On its website, the business promotes the multi-million-dollar initiative directly to consumers. Our results indicate that programs like this are attractive to job seekers, regardless of their self-transcendence or self-enhancement values, and are worthy of promotion as part of a recruitment strategy. In an article on employer branding, Ambler and Barrow argue that businesses are missing an important opportunity by focusing narrowly on consumers and investors, and by neglecting to build brand loyalty in the eyes of prospective and current employees [32].

Organizations can also make better use of new online tools, such as the CSRHub, to benchmark their progress on CSR and use this comparative information to attract high quality staff [33]. Currently, 18,958 companies from 143 countries have signed up to the CSRHub, indicating that this is a priority for many organizations. The program is voluntary, so the organizations are likely leaders in CSR implementation. Evaluation of the 'employees' category has a top-down focus by assessing, for example, "robust delivery (EEO-1) programs and training". Likewise, the subcategory 'compensation and benefits' also indicates a management-centric perspective by evaluating "the company's capacity to increase its workforce loyalty and productivity through rewarding, fair, and equal compensation and financial benefits" [33]. It appears that this reporting targets investors; however, employees might also be targeted if the reports were enhanced by an employee perspective.

Overall, the three key areas that organizations might emphasize when refining and marketing their brands are a positive social working environment, a commitment to positive environmental outcomes, and customer and societal well-being. Other research has identified a long list of factors that predict attractiveness, but our study shows these three features to be particularly important [1, 2].

### Limitations and future research

This study had several limitations that should be considered when interpreting our findings. First, our study relies on self-reported data provided by job seekers recruited from a non-probability sample. Although we employed a large, diverse national sample, findings cannot presume to be generalizable to the broader Australian population or to other countries. To evaluate the robustness of our findings, we recommend additional studies using a variety of samples, including those from other countries and cultures, recruited in ways other than through an online panel, and with more balanced gender distributions.

A second limitation is that our study focused on only six organization attributes, five derived from the core dimensions identified by Berthon et al. in their work developing the EAS, and one additional dimension related to corporate environmental responsibility [4]. The EAS encompasses a broad range of attributes, from pay to positive social interactions. However, other dimensions such as work/life balance, and the organization's image and familiarity to the applicant, which have been shown to be significantly associated with perceived organization attractiveness, were not included in our study. Previous research has shown that recruiter behaviours and characteristics of the recruitment processes significantly influence attraction [1, 2]. Future research should systematically examine a broader range of organization attributes, including those described above, to determine how they interact with applicant values to predict organization attraction.

Third, the six organization attributes investigated in this study were presented in the same order across presentations, with the economic value attribute always presented first, and the

environmental value attribute always presented last. This introduces the possibility of order effects, potentially increasing the impact of attributes presented early (primacy effect) or late in the scenarios (recency effect). To rule out order effects, we recommend future studies present attributes in a random order.

Fourth, the magnitude of difference between the high-low variants of the organization attributes used in our scenarios were not exactly equal. For example, whereas the social value attribute varied from negative and unpleasant (low) to positive and pleasant (high), the economic value attribute varied from average pay and conditions (low) to above average pay and conditions (high). Thus, the relative differences in the effect sizes for the Level 1 (organizational attribute) predictors should be interpreted with caution. Despite this limitation, it is worth noting that (1) all Level 1 predictors were statistically reliable predictors of job seekers' organization attraction judgements, and (2) the primary purpose of the study relates not to the relative strength of the Level 1 predictors, but the extent to which the direction and magnitude of these effects changed as a function of job seekers personal values. Nevertheless, future research should attempt to ensure high and low variants remain more commensurate across attributes.

Fifth, our study focused on job applicants' perceptions of organization attractiveness. Although perceived attractiveness is an important determinant of whether applicants will actually choose specific jobs, it is neither a necessary nor a sufficient cause. A meta-analytic review by Chapman et al. [1] found that organization attribute studies that used attractiveness as the primary dependent variable typically produced effect sizes that were nearly twice as large as studies that used job choice. Thus, further research is needed to determine whether the effects found in the current study for attractiveness ratings can be replicated using a behavior-based choice measure.

Finally, concerning individual difference factors, we focused on two types of personal values: self-transcendence and self-enhancement. Other personal characteristics may act as potential moderators and should be investigated in subsequent research. For example, a recent meta-analysis on individual-level differences and organization attraction found that applicant ability, personality, and experience were more important predictors of attraction than race, gender, and age [34]. A framework developed by Ambler and Barrow categorizes job and organization attributes based on functional, economic, and psychological benefits associated with employment [32]. This framework facilitates the systematic study of personal characteristics and psychological processes that might influence perceptions of organization attractiveness for both current and future employees.

## Conclusion

The study combined person-organization fit theory and a policy capturing methodology to determine (a) which workplace attributes are the strongest predictors of perceived organization attractiveness in a sample of Australian job seekers, and (b) whether the magnitude of these predictive effects varied as a function of job seekers' personal values. The three strongest drivers of perceived organization attractiveness were the provision of positive social environments, commitment to customer/societal well-being, and pro-environmental responsibility. These drivers were significantly more impactful than pay rates, opportunities for personal and career development, and stimulating/innovative work environments. We also found that personal values moderated the impact of workplace attributes and perceived attractiveness of organizations. In particular, job seekers with strong self-transcendence values and weak self-enhancement values were most sensitive to the absence of social, environmental, and application value, down-rating organizations that scored low on these attributes. Overall, our findings highlight the importance of understanding both workplace conditions and the values of job

seekers in the recruitment process. Non-collegial workplaces that undervalue customer, societal, and environmental outcomes are less attractive to job seekers, and this effect is particularly pronounced for those with who value "bigger than self" outcomes.

## Supporting information

**S1 Table. High and low descriptors for workplace attributes.**
(DOCX)

**S2 Table. Level 1 analysis: Workplace attributes predicting organization attractiveness.**
(DOCX)

**S3 Table. Level 2 analyses: Job seekers' personal values predicting level 1 effects.**
(DOCX)

**S1 Fig. Cross-level interactions between personal values and workplace attributes predicting job-seekers' perceptions of organizational attractiveness.**
(TIF)

**S2 Fig. Non-significant cross-level interactions between personal values and workplace attributes predicting job-seekers' perceptions of organizational attractiveness.**
(TIF)

## Author Contributions

**Conceptualization:** Carol L. Hicklenton, Donald W. Hine.

**Data curation:** Carol L. Hicklenton.

**Formal analysis:** Carol L. Hicklenton, Donald W. Hine.

**Methodology:** Donald W. Hine.

**Supervision:** Donald W. Hine.

**Writing – original draft:** Carol L. Hicklenton.

**Writing – review & editing:** Donald W. Hine, Aaron B. Driver, Natasha M. Loi.

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
