## [Decision Letter · Decision Letter 0]

25 Mar 2021

PONE-D-21-01887

How personal values shape job seeker preference: A policy capturing study

PLOS ONE

Dear Dr. Hicklenton,

Thank you for submitting your manuscript to PLOS ONE. After careful consideration, we feel that it has merit but does not fully meet PLOS ONE’s publication criteria as it currently stands. Therefore, we invite you to submit a revised version of the manuscript that addresses the points raised during the review process.

Based on my own reading of the paper and on very positive feedback from a referee I am happy to invite you to resubmit a revised version of the paper. Please make sure to address all the points raised by the referee and also the following: a) Please better specify the doi for data access. I succeeded using % instead of / but this is no longer working. Data used should be easily reachable. b) I think you should introduce more graphical analysis of all dimensions for both personal types, beyond that already present in Figure S1. For instance I was expecting a figure for Hypothesis 7. And please make sure to use the same scale in the axes for all comparable figures. c) Regarding the data: You should clarify how much did you pay the Qualtrix subject sample participants, specify the dropout rate, and compare your sample to the general Australian population. d) Finally, please acknowledge possible order effects due to presenting the attributes always in the same sequence.

We look forward to receiving your revised manuscript.

Kind regards,

Iván Barreda-Tarrazona, PhD

Academic Editor

PLOS ONE

Journal Requirements:

3. Please provide additional details regarding participant consent.

In the ethics statement in the Methods and online submission information, please ensure that you have specified (i) whether consent was informed and (ii) what type you obtained (for instance, written or verbal, and if verbal, how it was documented and witnessed). If your study included minors, state whether you obtained consent from parents or guardians. If the need for consent was waived by the ethics committee, please include this information.

4. Please upload a copy of Figure 1, to which you refer in your text on page 16. If the figure is no longer to be included as part of the submission please remove all reference to it within the text.

5. Please include a copy of Tables 1, 2 and 3 which you refer to in your text on pages 14 and 16.

Reviewers' comments:

Reviewer's Responses to Questions

**Comments to the Author**

1. Is the manuscript technically sound, and do the data support the conclusions?

Reviewer #1: Yes

2. Has the statistical analysis been performed appropriately and rigorously? 

Reviewer #1: Yes

3. Have the authors made all data underlying the findings in their manuscript fully available?

Reviewer #1: Yes

4. Is the manuscript presented in an intelligible fashion and written in standard English?

Reviewer #1: Yes

5. Review Comments to the Author

Reviewer #1: I enjoyed reading this paper on PO Fit and Organizational Attractiveness. This is ground that has been well-covered but the authors carefully outline how this paper adds to the literature. First, it does not utilize a student sample, which is an important differentiator. Second, it uses a well regarded instrument to measure individual values that adds credibility and generalizability to the paper.

I have a few comments:

-this is a very well written paper that clearly explicates the context, the study and the findings. I found it refreshing to read a manuscript that was carefully edited and so clearly written

-there are a number of issues that arise in this topic that the authors need to more forthrightly address, at least to quiet the critics. For example, org attractiveness does not necessarily translate to applying for positions at a firm. What do we know about the connection between an espoused policy and likelihood of taking action on it?

-please address social desirability bias. How can we understand how the study participants might have adapted their answers to demonstrate that they had sound and socially acceptable values?

-I found the language for the "low" condition in the scenarios to be quite negative and off putting. So rather than just saying that these factors are not really present in the company, it seemed that the company didn't care about those factors. It seemed overly negative

-can you explain more fully the impact of having only 6 individuals see the same scenario (400/64). Does the large number of scenarios in comparison to the sample size in any way inhibit your inferences? Please address that.

-in the practical implication section, you don't specifically address how firms can use the PO fit aspect of org attractiveness to reach out to the best fitting prospective applicants. Can you provide more information about these implications, given that the paper is focused on that issue? You found that certain attributes are simply more attractive, but how can we target the best fitting applicants. I believe that the implications section of the paper needs more work to provide interesting and potentially impactful ideas.

Thanks for the opportunity to review your paper.

6. PLOS authors have the option to publish the peer review history of their article (what does this mean?). If published, this will include your full peer review and any attached files.

Reviewer #1: No

---

## [Author Response · Author response to Decision Letter 0]

30 Jun 2021

Reviewer #1 comments

1. Organization attractiveness does not necessarily translate to applying for positions at a firm. What do we know about the connection between an espoused policy and likelihood of taking action on it?

Response: A paragraph discussing this has been added to the discussion section (600/26).

2. Address social desirability bias; How can we understand how the study participants might have adapted their answers to demonstrate that they had sound and socially acceptable values?

Response: Given that the survey employed an anonymous response format pressure to provide socially desirable responses should have be relatively low. In addition, to the extent that social desirability was operating, it is worth noting that this likely mimics virtue signalling in people’s real world choices, in which jobs are sometimes chosen to enhance social status and reputation concerns. 

3. The reviewer found the language for the "low" condition in the scenarios to be quite negative and off putting. The reviewer suggests than saying these factors are not really present in the company or that the company didn't care about those factors might be overly negative.

Response: We have added a sentence to the Limitations and future research section to address the point you raise (588/26).

4. You asked for a fuller explanation of the impact of having only 6 individuals see the same scenario (400/64). Does the large number of scenarios in comparison to the sample size in any way inhibit your inferences? 

Response; Although it is correct that only 6 participants would have seen exactly the same full scenario, 400 participants would have viewed each level (low vs high) of all of our organization attribute variables. Given that we were not investigating all possible interactions between all of the organisation attributes, the statistical power was high for all main effects and the limited number of cross-level interactions conducted. 

In terms of the potential for biased estimates MLM, Maas and Jox (2005) note that in MLM the major contributor to bias/inaccuracy is the higher level sample size – in the case of our study Level 2. In their paper, they ran a series of simulations in which they varied Level 1 sample size, Level 2 sample size, and magnitude of intraclass correlations. They found that only simulations with small Level 2 samples (consisting of 50 or fewer cases) produced biased Level 2 standard errors. All other simulations, including those with Level 1 sample sizes as small as 5, produced accurate and unbiased regression coefficients, variance components and standard errors at Level 1 and Level 2. 

We cite this paper and describe its findings on p. 16 of the revised manuscript.

Maas, C. J. M., & Hox, J. J. (2005). Sufficient sample sizes for multilevel modeling. Methodology, 1(3), 86-92. doi: 10.1027/1614-1881.1.3.86

5. Include in the practical implication section, specifically how firms can use the PO fit aspect of org attractiveness to reach out to the best fitting prospective applicants. Provide more information about these implications, given the paper is focused on that issue. You found that certain attributes are simply more attractive, but how can we target the best fitting applicants. I believe that the implications section of the paper needs more work to provide interesting and potentially impactful ideas.

Response; The practical implications section has been expanded with ideas on how to target the best fitting applicants and why this is important.

Additional requirements

1. Ensure the manuscript meets PLOS ONE's style requirements, including those for file naming.

Response; We’ve checked our submission meets PLOS ONE’s style and file naming requirements. 

2. Review the reference list to ensure that it is complete and correct. 

Response; We’ve checked to ensure the reference list is complete and correct. 

3. Provide additional details regarding participant consent - was consent informed, were any participants minors, type of consent obtained.

Response; Additional details about consent have been added to the Procedure section (243/11. 

4. Upload a copy of Figure 1 referred to on page 16 of the original submission. If the figure is no longer to be included as part of the submission, please remove all reference to it within the text.

Response; S1 Figure showing significant interactions has been updated. A second figure (S2 Figure) for non-significant interactions has also been added. If possible, we would prefer to include S2 Figure as supplementary material with a note in the article that refers to the second figure. 

5. Include a copy of Tables 1, 2 and 3 referred to in on pages 14 and 16 of the original submission.

Response; These tables are included in the resubmission

6. Consider introducing more graphical analysis of all dimensions for both personal types, beyond that already present in Figure S1. The editor expected to see a figure for Hypothesis 7.

Response; A second figure for non-significant interactions has been added to the submission as supplementary material with a note in the article that refers to the second figure.

7. Consider using the same scale in the axes for all comparable figures.

Response; We have changed the figures so all axes use the same scale. 

8. Specify how much we paid the Qualtrics subject sample participants and the dropout rate.

Response; This information has been added to the Procedure section (245/11).

9. Compare our sample to the general Australian population

Response; In the Participants section, we now compare our sample to the general Australian population (228/11). 

10. Acknowledge possible order effects due to presenting the attributes always in the same sequence.

Response; The risk of order effects has been included in a paragraph in the Discussion section (582/25).

---

## [Editor Report · Decision Letter 1]

1 Jul 2021

How personal values shape job seeker preference: A policy capturing study

PONE-D-21-01887R1

Dear Dr. Hicklenton,

We’re pleased to inform you that your manuscript has been judged scientifically suitable for publication and will be formally accepted for publication once it meets all outstanding technical requirements.

Kind regards,

Iván Barreda-Tarrazona, PhD

Academic Editor

PLOS ONE

Additional Editor Comments (optional): You have successfully addressed all the concerns raised in the revision process.
---

## [Editor Report · Acceptance letter]

21 Jul 2021

PONE-D-21-01887R1 

How personal values shape job seeker preference: A policy capturing study 

Dear Dr. Hicklenton:

I'm pleased to inform you that your manuscript has been deemed suitable for publication in PLOS ONE. Congratulations! Your manuscript is now with our production department. 

Kind regards, 

on behalf of

Dr. Iván Barreda-Tarrazona 

Academic Editor

PLOS ONE